# Lipocalin-2 Deficiency Reduces Oxidative Stress and Neuroinflammation and Results in Attenuation of Kainic Acid-Induced Hippocampal Cell Death

**DOI:** 10.3390/antiox10010100

**Published:** 2021-01-12

**Authors:** Hyun Joo Shin, Eun Ae Jeong, Jong Youl Lee, Hyeong Seok An, Hye Min Jang, Yu Jeong Ahn, Jaewoong Lee, Kyung Eun Kim, Gu Seob Roh

**Affiliations:** Department of Anatomy and Convergence Medical Science, Bio Anti-aging Medical Research Center, Institute of Health Sciences, College of Medicine, Gyeongsang National University, Jinju 52777, Gyeongnam, Korea; k4900@hanmail.net (H.J.S.); jeasky44@naver.com (E.A.J.); jyv7874v@naver.com (J.Y.L.); gudtjr5287@hanmail.net (H.S.A.); gpals759@naver.com (H.M.J.); ahnujung@naver.com (Y.J.A.); woongs1111@gmail.com (J.L.); kke-jws@hanmail.net (K.E.K.)

**Keywords:** kainic acid, lipocalin-2, oxidative stress, neuroinflammation, hippocampus

## Abstract

The hippocampal cell death that follows kainic acid (KA)-induced seizures is associated with blood–brain barrier (BBB) leakage and oxidative stress. Lipocalin-2 (LCN2) is an iron-trafficking protein which contributes to both oxidative stress and inflammation. However, LCN2′s role in KA-induced hippocampal cell death is not clear. Here, we examine the effect of blocking LCN2 genetically on neuroinflammation and oxidative stress in KA-induced neuronal death. LCN2 deficiency reduced neuronal cell death and BBB leakage in the KA-treated hippocampus. In addition to LCN2 upregulation in the KA-treated hippocampus, circulating LCN2 levels were significantly increased in KA-treated wild-type (WT) mice. In LCN2 knockout mice, we found that the expressions of neutrophil markers myeloperoxidase and neutrophil elastase were decreased compared to their expressions in WT mice following KA treatment. Furthermore, LCN2 deficiency also attenuated KA-induced iron overload and oxidative stress in the hippocampus. These findings indicate that LCN2 may play an important role in iron-related oxidative stress and neuroinflammation in KA-induced hippocampal cell death.

## 1. Introduction

Kainic acid (KA) is an analog of the excitatory amino acid transmitter glutamate, and experimental seizures induced by KA are accompanied by neuronal cell death associated with blood–brain barrier (BBB) leakage and oxidative stress [1,2]. After such KA-induced neuronal injury, increased iron levels were observed in the rat hippocampus as well as higher intracellular calcium levels [3]. Increased iron levels, especially ferrous iron (Fe^2+^) can exceed the cell’s detoxification systems and enhance the conversion of H_2_O_2_ to ∙OH, favoring greater turnover in the Haber–Weiss cycle and resulting in an amplification of oxidative stress [4]. Increased oxidative stress has been regarded as an important underlying cause for seizure-induced neuronal damage [5,6] and some antioxidants are present at high concentrations in astrocytes, protecting the surrounding cells from oxidative stress-induced cell death [7,8]. Moreover, previous studies have shown that c-Jun N-terminal kinase-signaling pathway plays an important role in the process of neuronal death in experimental epilepsy model [9,10].

Lipocalin-2 (LCN2), also known as neutrophil gelatinase-associated lipocalin, is an acute-phase brain-injury protein produced by the choroid plexus that induces several chemokines to participate in the inflammatory response [11,12]. It also has additional roles in a variety of processes, including iron-loaded bacterial siderophores, mammalian iron metabolism, anti- and pro-inflammatory responses, and both cell migration and differentiation to defend against certain bacterial infections [13,14,15]. LCN2 is known to be released from the choroid plexus in response to peripheral lipopolysaccharide (LPS) administration [11] and after excitotoxic brain injury [16]. Moreover, several reports have shown that LCN2 contributes to neuronal cell death in animal brain-injury models, such as LPS-induced neuroinflammation, encephalomyelitis, and cerebral ischemia [15,17,18]. Despite these investigations, the role of LCN2 in KA-induced neuronal cell death is not well understood.

In the present study, we examine the effect of LCN2 deficiency on iron-related oxidative stress and neuroinflammation in the KA-treated hippocampus to determine the role of LCN2 in KA-induced hippocampal cell death. Our findings indicate that LCN2 may play a critical role in iron-mediated oxidative stress and neuroinflammation, and that blocking LCN2 genetically may protect against KA-induced hippocampal cell death.

## 2. Materials and Methods

### 2.1. Animals

Male C57BL/6J mice were purchased from Central Laboratory Animal, Inc. (Seoul, South Korea), and LCN2 (−/−) mice were purchased from The Jackson Laboratory (Bar Harbor, ME, USA). Wild-type (WT) and LCN2 (−/−) mice were back-crossed on the C57BL/6J background for 8 to 10 generations to produce homogeneous animals, with no background effects on phenotypes. The absence of LCN2 was confirmed by polymerase chain reaction (PCR) analysis of genomic DNA. All animal experiments were performed in accordance with approved animal protocols and guidelines established by the Animal Care Committee at Gyeongsang National University (No. GNU-190701-M0033). Eight-week-old WT and LCN2 (−/−) mice were used for this study.

### 2.2. The KA-induced Seizure Model

Mice were treated with an intraperitoneal injection (20 mg/kg) of KA (Abcam, Cambridge, UK), as previously reported [1]. Mice in the control groups (WT-CTL and LCN2 (−/−)-CTL; n = 10 mice per group) were injected with 0.9% normal saline. WT and LCN2 (−/−) mice (n = 10 mice per group) were sacrificed at 6 h and 24 h after KA treatment.

### 2.3. Tissue Preparation and Cresyl Violet Staining

For tissue analyses, mice (n = 4 mice per group) were anesthetized with zoletil (5 mg/kg; Virbac Laboratories, Carros, France) and then perfused transcardially with saline followed by 4% paraformaldehyde in 0.1 M phosphate buffered saline (PBS). Six hours after post fixation in the same fixative, the brains were sequentially immersed in 0.1 M PBS containing 15% sucrose and then in PBS containing 30% sucrose at 4 °C until they sank. After freezing, the frozen brains were cut into 40-μm thick coronal sections. To inspect general histology and pyknotic nucleus, the sections were stained with Cresyl violet, visualized with a BX51 light microscope (Olympus, Tokyo, Japan), and digital images were captured.

### 2.4. Terminal Deoxynucleotidyl Transferase dUTP Nick End Labeling (TUNEL) Assay

TUNEL analyses were performed to measure the degree of apoptosis in tissue using an in situ cell death detection kit (Roche Molecular Biochemicals, Mannheim, Germany) according to the manufacturer’s protocol. Nuclei were counterstained with 4′,6-diamidino-2-phenylindole (DAPI; Invitrogen, Waltham, MA, USA). Fluorescence was visualized using a BX51-DSU microscope (Olympus), and digital images were captured. TUNEL-positive cells were counted in the CA3 region (200 × 200 µm) in three sections (n = 4 mice per group) using ImageJ software (Version 1.52a, NIH, Bethesda, MD, USA).

### 2.5. LCN2 Enzyme-Linked Immunosorbent Assay (ELISA)

Blood samples (n = 5 mice per group) were taken from the left ventricle and centrifuged. Amounts of LCN2 in serum were quantified using an ELISA kit (R&D Systems, Minneapolis, MN, USA) according to the manufacturer’s instructions.

### 2.6. Immunofluorescence

Free-floating brain tissue sections were incubated with primary antibodies (Table 1) overnight at 4 °C. After washing, the sections were incubated with Alexa Fluor 488-, 594- or 680-conjugated secondary antibodies (Invitrogen Life Technologies, Carlsbad, CA, USA). Nuclei were counterstained with DAPI (Invitrogen). Fluorescence was visualized using a BX51-DSU microscope (Olympus), and digital images were captured.

### 2.7. Diaminobenzidine (DAB)-Enhanced Perls’ Iron Staining

Brain sections from WT and LCN2 (−/−) mice were used to determine iron levels. For detection of ferric iron, DAB-enhanced Perls’ iron staining was performed, as described previously [19]. Briefly, brain sections were incubated in Perls’ solution (Abcam) for 30 min, followed by incubation in 0.05% DAB substrate kit (Vector Laboratories, Burlingame, CA, USA) for 20 min. Then, 1% H_2_O_2_ was added, and samples were incubated for 30 min. Sections were then washed, covered with mounting medium, and visualized using a BX51 light microscope (Olympus). Quantification of Perls’ stain in CA3 was carried out by measuring staining intensity in brain sections from WT and LCN2 (−/−) mice using ImageJ software (Version 1.52).

### 2.8. Western Blotting

Frozen hippocampi (n = 6 mice per group) were homogenized in lysis buffer (Thermo Fisher Scientific, Waltham, MA, USA) for the immunodetection of proteins. To obtain the nuclear fraction, we used the NE-PER Nuclear and Cytoplasmic Extraction Kit (Pierce, Rockford, IL, USA). Protein concentrations were determined using a Bio-Rad protein assay, and samples were stored at −80 °C until use. The primary antibodies used are shown in Table 1. β-actin or p84 were used as loading control to normalize protein levels in total or nuclear fraction, respectively. Additionally, IgG was used as loading control in serum fraction. Each protein was visualized using an enhanced chemiluminescence substrate (Pierce). The relative amounts of targeted proteins were quantified from band intensities using the Multi-Gauge V 3.0 image analysis program (Fujifilm, Tokyo, Japan).

### 2.9. Statistical Analyses

Statistical analyses were performed using PRISM 7.0 (GraphPad Software, Inc., San Diego, CA, USA). Two-group differences were determined by Student *t*-tests. For the time course data, differences were first determined using a two-way analysis of variance (ANOVA), followed by Tukey’s post-hoc comparisons. All values are expressed as mean ± SEM. *p* < 0.05 was considered significant.

## 3. Results

### 3.1. Effect of LCN2 Deficiency on KA-Induced Neuronal Cell Death in the Mouse Hippocampus

To examine whether LCN2 deficiency affected KA-induced neuronal cell death, WT mice and LCN2 (−/−) mice were sacrificed at 6 h or 24 h after KA injection. Cresyl violet staining showed that WT and LCN2 (−/−) mice had pyknotic nuclei (typically found in apoptotic cells) in the hippocampal CA3 region 24 h after KA injection (Figure 1A). To evaluate if the observed cell death was apoptotic, a TUNEL assay was performed (Figure 1B). The proportion of TUNEL-positive cells was decreased in the hippocampi of LCN2 (−/−) mice (4.177 ± 0.28%) compared to WT mice (16.15 ± 3.64%) 24 h post-KA injection (*p* < 0.05). Western blot analysis showed that the level of Rab5, an early endosome marker, in WT mice increased in the hippocampus 6 h after KA injection compared to its level in LCN2 (−/−) mice (Figure 1C). In addition, the expression of lysosomal-associated membrane protein 1 (LAMP1), a late endosome/lysosomal marker, increased in the hippocampi of WT mice 6 h and 24 h after KA treatment, but there was no significant change in its expression in KA-treated LCN2 (−/−) mice (Figure 1D).

### 3.2. Effect of LCN2 Deficiency on BBB Leakage in the KA-Treated Hippocampus

We examined whether LCN2 deficiency affected KA-induced BBB leakage in the mouse hippocampus. LCN2 deletion attenuated the KA-induced expressions of the glial water-channel aquaporin 4 (AQP4) and the tight junction protein claudin-5 in the hippocampus beginning 6 h post-KA injection (Figure 2A,B). Vascular cell adhesion molecule 1 (VCAM-1) immunofluorescence labeling was observed in the blood vessels of the CA3 hippocampus in WT mice 24 h after KA treatment, but its labeling in LCN2 (−/−) mice remained unchanged (Figure 2C). In both WT and LCN2 (−/−) mice, CD68-positive macrophages were detected in the CA3 hippocampus 6 h and 24 h post-KA injection (Figure 2C). Neutrophils are known to produce cytotoxic inflammatory mediators in the brain [20], and we found that the neutrophil marker proteins myeloperoxidase (MPO) and neutrophil elastase (NE) were both increased in KA-treated hippocampi compared to hippocampi of LCN2 (−/−) mice 6 h after KA injection (Figure 2D,E). Taken together, these findings indicate that LCN2 deficiency could inhibit neutrophil infiltration through KA-induced BBB breakdown.

### 3.3. Effect of LCN2 Deficiency on Neuroinflammation in the KA-Treated Hippocampus

We found that KA-induced GFAP protein levels in the hippocampus were significantly increased 24 h post-KA injection compared to LCN2 (−/−) mice (Figure 3A). We also examined the levels of neuroinflammation-related proteins [interleukin-6 (IL-6), nuclear factor-kappaBp65 (NF-κBp65), and cyclooxygenase-2 (COX-2)] in both KA-treated WT and LCN2 (−/−) mice. LCN2 deletion attenuated the expressions of KA-induced nuclear NF-κBp65, IL-6, and COX-2 in the hippocampus (Figure 3B–D).

### 3.4. KA Treatment Increased Circulating and Hippocampal LCN2 Levels

Using ELISA and Western blot analyses, we found that circulating LCN2 levels were increased in KA-treated WT mice 6 h and 24 h post-KA injections (Figure 4A,B). In addition, we observed LCN2-positive cells in the third ventricle of WT mice 6 h and 24 h after KA treatment (Figure 4C), indicating that LCN2 could be secreted due to KA-induced BBB leakage and neuroinflammation. Next, we observed that hippocampal LCN2 expression was increased in WT mice 6 h and 24 h after KA administration (Figure 4D). Interestingly, 24p3R protein (the LCN2 receptor) was increased in KA-treated WT mice at 6 h, then decreased by 24 h, but there was no change in LCN2 (−/−) mice (Figure 4E). Using immunofluorescence staining, we observed that LCN2-positive cells co-localized with GFAP-positive astrocytes around dead neurons in KA-treated CA3 hippocampal regions at 24 h, but 24p3R-positive cells were abundantly observed in hippocampal neurons (Figure 4F).

### 3.5. Effect of LCN2 Deficiency on Iron Accumulation and Oxidative Stress in KA-Treated Mouse Hippocampus

LCN2 has emerged as a critical iron-regulatory protein during central nervous system inflammation [21], so we examined whether LCN2 deficiency affected KA-induced iron accumulation and oxidative stress. Using DAB-enhanced Perls’ iron staining, we found that iron, predominantly localized to neurons, increased in KA-treated WT mice at 24 h compared to iron staining in LCN2 (−/−) mice (Figure 5A,B). We also examined the expression of iron-related proteins, such as iron storage protein ferritin and a ferroxidase ceruloplasmin (Figure 5C,D). KA-induced hippocampal ferritin expression was significantly increased in WT mice 24 h post-KA administration (Figure 5C). In particular, we found that LCN2 deletion inhibited KA-induced ceruloplasmin expression beginning 6 h after KA treatment (Figure 5D). Finally, we examined the levels of the oxidative stress-related proteins inducible nitric oxide synthase (iNOS) and heme oxygenase-1 (HO-1) in KA-treated mice, and found that the genetic blocking of LCN2 inhibited KA-induced hippocampal iNOS expression in WT mice (Figure 5E). The activation of HO-1 in response to brain injury may be based on the induction of genes associated with the antioxidant response factor [22], and we found that KA-induced HO-1 expression was significantly increased at 24 h compared to its expression in LCN2 (−/−) mice (Figure 5F). Furthermore, we observed that LCN2-positive astrocytes co-localized with HO-1-positive cells around dead neurons in KA-treated CA3 hippocampal regions at 24 h (Figure 5G). These findings indicate that KA-induced iron overload and oxidative stress may be inhibited by LCN2 deletion.

## 4. Discussion

In this study, we focused on the role of iron-transporting LCN2 in the KA-treated hippocampus, and for the first time have demonstrated that KA simultaneously induces not only hippocampal LCN2, but also circulating LCN2 at 6 h after KA treatment. Notably, LCN2-deficient mice exhibited reduced KA-induced neuronal cell death, BBB leakage, and neuroinflammation by minimizing iron-binding protein LCN2-mediated oxidative stress.

KA-induced neuronal death has been closely linked to endoplasmic reticulum stress and mitochondrial apoptosis [23], and Rab5, an early endosomal marker, is translocated to mitochondria in response to apoptotic signals by oxidative stress after brain injury [24]. Previous studies have shown that Rab5 activation mediates neuronal apoptosis caused by a familial Alzheimer’s disease (AD) mutant of amyloid precursor protein (APP) [25] and that LAMP1 plays a role in KA-mediated apoptotic neuronal death [26]. In addition, the accumulation of β-APP via lysosomal dysfunction has been demonstrated in both AD and cerebral ischemia [27]. We found significantly attenuated KA-induced hippocampal Rab5 and LAMP1 expressions in LCN2 (−/−) mice, indicating that LCN2 may play a critical role in Rab5 and LAMP1-mediated apoptosis after KA treatment.

In a previous study, we demonstrated that adiponectin pretreatment (which preserves the integrity of the BBB) has a neuroprotective effect against KA-induced seizures by reducing VEGF, eNOS, NF-κBp65 expression [28]. In addition, we have reported that tonicity-responsive enhancer binding protein haploinsufficiency had a neuroprotective effect against KA-induced seizures by reducing NF-κB-mediated inflammation, and by reducing VEGF and AQP4 associated with BBB leakage [1]. Treatment with KA is known to activate many pro-inflammatory cytokines such as IL-1β, tumor necrosis factor (TNF)-α, and iNOS, ultimately leading to astrocyte activation and induction of neuronal cell apoptosis in the brain [29]. In addition, hippocampal COX-2 expression has also been shown to significantly increase 6 h and 24 h after KA treatment [1,30]. In the present study, we found that LCN2 (−/−) mice had reduced hippocampal AQP4, claudin-5, and VCAM-1 expressions after systemic KA administration. Furthermore, LCN2 (−/−) mice exhibited a significant reduction in neuroinflammation as well as expressions of IL-6 and NF-κBp65-mediated COX-2 in the KA-treated hippocampus. Taken together, these findings indicate that LCN2 deficiency may protect against neuronal death through attenuation of KA-induced BBB leakage and neuroinflammation.

It has been shown that BBB leakage leads to the accumulation of circulating mediators in the brain, leading to increased excitotoxicity [31]. Neutrophils are the most abundant leukocytes in blood [32], and neutrophil entry into brain tissue increases under pathological conditions, such as infections, trauma, ischemia, and hemorrhage [33]. MPO is a key enzyme expressed in blood-borne neutrophils, macrophages and monocytes, and has been used as a marker for inflammation after brain injuries [34]. Ablation of MPO has been reported to alleviate features of Parkinson’s disease in mice [35], and resveratrol has been shown to confer protection against rotenone-induced neurotoxicity by modulating MPO levels in glial cells [36]. NE, a myeloid-specific serine protease [37], is known to promote degradation of the extracellular matrix and to trigger pathways leading to cell death [38,39]. Here, we found that LCN2 deficiency significantly reduced the KA-induced neutrophil markers MPO and NE in the hippocampus. These data suggest that LCN2, as a pro-inflammatory mediator, may promote neutrophil infiltration across a leaking BBB after KA treatment.

LCN2 is well known as both an acute-phase protein and a siderophore-binding protein. Chia et al. [16] demonstrated that LCN2 is expressed in a variety of normal rat brain regions, such as the olfactory bulb, cerebellar cortex, different areas of the brainstem, and in epithelial cells of the choroid plexus. In addition, several studies have shown that the levels of hippocampal LCN2 expression increase in reactive astrocytes 24 h after treatment with either LPS or KA [15,16]. Similar to the present findings, our previous work also showed that hippocampal LCN2 increased 6 h after KA treatment in mice [40], and LCN2 was considered to rapidly influence the hippocampus after KA treatment. The present study also confirmed that LCN2 expression was observed in reactive astrocytes in the hippocampal CA3 region 6 h and 24 h after KA administration as parallel findings. In addition, other studies have shown that LCN2 protein levels in the blood increase both with age and with mild cognitive impairment, and are also increased in human post-mortem brain specimens with different diseases of the central nervous system, including AD, Parkinson’s disease, and multiple sclerosis [41,42,43,44]. Serum levels of LCN2 have also been shown to progressively increase in a rat middle cerebral artery occlusion model, and that this increase affected endothelial function as an important part of BBB-integrity maintenance [45]. Importantly, and for the first time, we have demonstrated that circulating LCN2 levels increase in both the serum and the choroid plexus 6 h and 24 h after KA treatment in WT mice. Our findings suggest that increased circulating LCN2 levels could be released from the KA-damaged hippocampus and cross the BBB.

Iron is an essential element of the brain, and a potent source for reactive oxidative stress [46]. The dysregulation of iron metabolism associated with cellular damage and oxidative stress has been reported to be common in several neurodegenerative disorders, including AD and Huntington’s disease [47,48]. In addition to high calcium levels in a KA-induced model for excitotoxic injury, a progressive increase in the iron concentration in rat hippocampus has been reported [3], and the oxidative stress condition of cerebral ischemia also resulted in increased expressions of the iron storage protein ferritin and ferroxidase ceruloplasmin [49]. We also found that the ferritin and ceruloplasmin expressions in the hippocampi of KA-treated mice were reduced by LCN2 deletion. Nitric oxide, produced by iNOS, is known to cause neuronal damage and apoptotic cell death after KA administration [50], and LCN2 has been shown to produce a damaging effect after cerebral ischemia by inducing high iNOS expression in astrocytes both in vivo and in vitro [51]. Consistently, we found that iNOS expression increased in a time-dependent manner in KA-treated WT mice, but remained at baseline in LCN2 (−/−) mice. Oxidative stress-related HO-1 expression has been shown to be upregulated within reactive glia and infiltrating macrophages as a function of the injury stimuli [52], and early traumatic brain-injury models have shown that a rapid induction of HO-1 is associated with cytotoxicity [53]. HO-1 is also responsible for the oxidation of heme groups, leading to the generation of biliverdin, carbon monoxide, and the release of Fe^2+^ [54]. Therefore, LCN2 has been suggested to increase as a result of iron overload produced by overwhelming HO-1-induced heme lysis, and has been shown to be co-localized with HO-1 one day after traumatic brain injury in a subpopulation of reactive glia in injured brain regions [52]. Accordingly, we found that HO-1 was increased in the KA-treated hippocampi of WT mice compared to its level in KA-treated LCN2 (−/−) mice 24 h after KA administration. The tissue from LCN2 deficient mice showed fewer immunostained HO-1-positive cells co-localized with LCN2-positive astrocytes in the KA-treated hippocampi of WT mice 24 h after KA administration. Taken together, these data suggest that LCN2-mediated iron accumulation may cause oxidative stress in the KA-treated hippocampus.

## 5. Conclusions

In conclusion, our findings suggest that the genetic blocking of LCN2 has a neuroprotective effect against KA-induced neuronal death through a reduction in iron-related oxidative stress. We therefore suggest that LCN2 may play an important role in neuroinflammation and oxidative stress induced by KA administration, and that LCN2 may be a potential therapeutic target against seizure-induced neuronal cell death and for other neurodegenerative diseases.

## Figures and Tables

**Figure 1 antioxidants-10-00100-f001:**
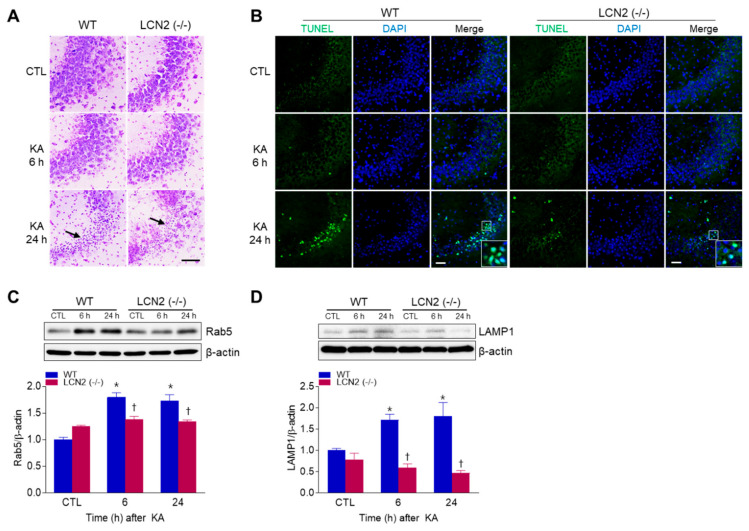
Effects of lipocalin-2 (LCN2) deficiency on hippocampal cell death in kainic acid (KA)-treated mice. (**A**) Cresyl violet-stained sections from KA-treated wild-type (WT) and LCN2 (−/−) mice. Arrows indicate CA3 regions that appear with pyknotic nuclei. (**B**) Representative immunofluorescence images (×200) of TUNEL staining (green) in hippocampal CA3 regions, counterstained with DAPI (blue). Western blot analysis and quantitative expressions of Rab5 (**C**) and LAMP1 (**D**) in the hippocampus. β-actin was used as a loading control. Data are shown as mean ± SEM. * *p* < 0.05 vs. WT-CTL. † *p* < 0.05 vs. WT + KA 6 h or WT + KA 24 h.

**Figure 2 antioxidants-10-00100-f002:**
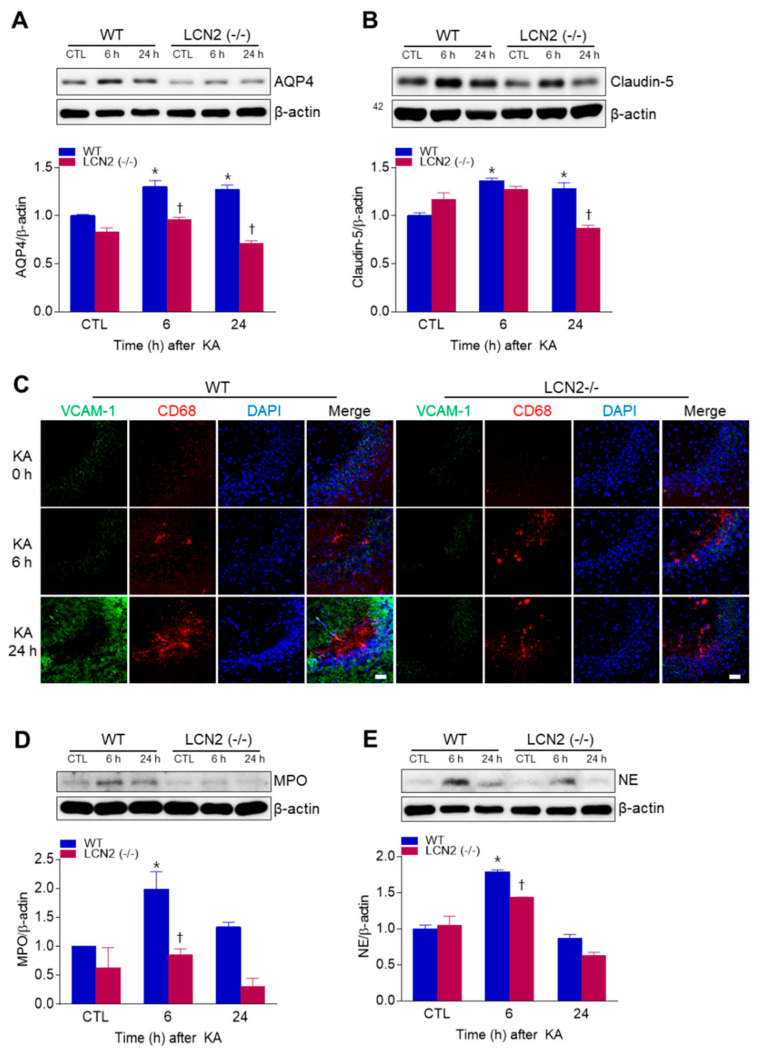
Effects of LCN2 deficiency on blood–brain barrier (BBB) leakage in hippocampi of KA-treated mice. (A and B) Western blot analysis and quantitative expressions of AQP4 (**A**) and claudin-5 (**B**) in the hippocampus. (**C**) Representative immunofluorescence images (×200) of hippocampal CA3 regions showing VCAM-1 (green), CD68 (red), and DAPI (blue) staining of nuclei. Scale bar = 25 µm. (D and E) Western blot analysis and quantitative expressions of MPO (**D**) and NE (E) in the hippocampus. β-actin was used as a loading control. Data are shown as mean ± SEM. * *p* < 0.05 vs. WT-CTL. † *p* < 0.05 vs. WT+KA 6 h or WT + KA 24 h.

**Figure 3 antioxidants-10-00100-f003:**
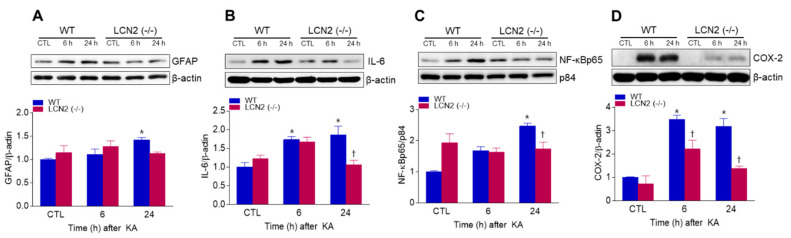
Effects of LCN2 deficiency on neuroinflammation in hippocampi of KA-treated mice. (**A**–**D**) Western blot analysis and quantitative expression of GFAP (**A**), IL-6 (**B**), NF-κBp65 (**C**), and COX-2 (**D**) in the hippocampus. β-actin and p84 were used as loading controls. Data are shown as mean ± SEM. * *p* < 0.05 vs. WT-CTL. † *p* < 0.05 vs. WT + KA 6 h or WT + KA 24 h.

**Figure 4 antioxidants-10-00100-f004:**
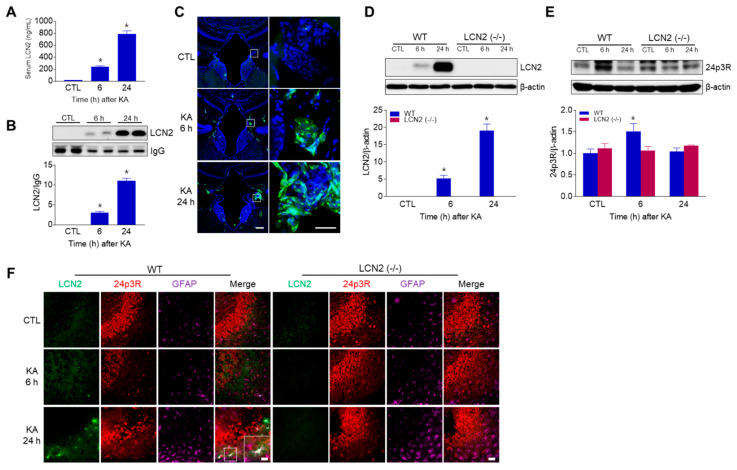
LCN2 expressions in serum, choroid plexus, and hippocampus after i.p. administration of KA. (**A**) ELISA values for serum LCN2 levels in WT mice. Data are shown as mean ± SEM. * *p* < 0.05 vs. WT-CTL. (**B**) Western blot analysis and quantification for serum LCN2 expression in the hippocampus. IgG was used as a loading control. Data are shown as mean ± SEM. * *p* < 0.05 vs. WT-CTL. (**C**) Representative immunofluorescence images (×100) of the choroid plexus of the third ventricle showing LCN2 (green) and DAPI (blue) staining of nuclei. Scale bar = 50 µm (inset, 25 µm). Western blot analysis and quantitative expressions of LCN2 (**D**) and 24p3R (**E**) in the hippocampus. β-actin was used as a loading control. Data are shown as mean ± SEM. * *p* < 0.05 vs. WT-CTL. (**F**) Representative immunofluorescence images (×200) of hippocampal CA3 regions showing LCN2 (green), 24p3R (red), and GFAP (purple) staining. Scale bar = 25 µm.

**Figure 5 antioxidants-10-00100-f005:**
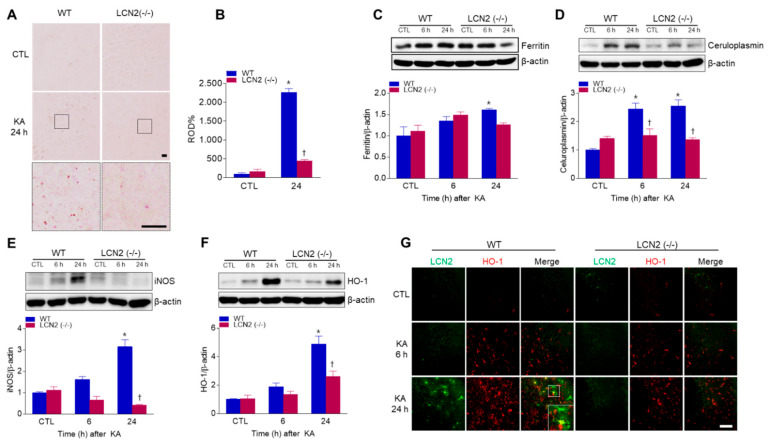
Effects of LCN2 deficiency on iron accumulation and oxidative stress after KA administration. (**A**) Histological staining (×400) for iron with DAB-enhanced Perls’ staining in the hippocampal CA3 regions of WT and LCN2 (−/−) mice. Scale bar = 10 µm (inset, 10 µm). (**B**) Relative optical density (ROD) measurements (%) from DAB-enhanced Perls’ iron staining. Western blot analysis and quantitative expression of ferritin (**C**), ceruloplasmin (**D**), iNOS (**E**), and HO-1 (**F**) in the hippocampus. (**G**) Representative immunofluorescence images (×200) of hippocampal CA3 regions showing LCN2 (green) and HO-1 (red) staining. Scale bar = 25 µm. β-actin was used as a loading control. Data are shown as mean ± SEM. * *p* < 0.05 vs. WT-CTL. † *p* < 0.05 vs. WT + KA 6 h or WT + KA 24 h.

**Table 1 antioxidants-10-00100-t001:** List of primary antibodies.

Classified Markers	Antibody	Company	Catalog No.	Dilution(s)	Applications	Source
Endosome/lysosome	Rab5	Santa Cruz	sc-46692	1:1000	WB	Mouse
	LAMP1	Abcam	ab24170	1:1000	WB	Rabbit
BBB leakage	AQP4	Abcam	ab46182	1:1000	WB	Rabbit
	Claudin-5	Thermo	35-2500	1:1000	WB	Mouse
	VCAM-1	Abcam	ab134047	1:100	IF	Rabbit
	CD68	Santa Cruz	sc-20060	1:100	IF	Mouse
	MPO	Abclonal	A1374	1:1000	WB	Rabbit
	NE	Abcam	ab68672	1:1000	WB	Rabbit
Neuroinflammation	GFAP	Sigma	G3893	1:5000, 1:500	WB, IF	Mouse
	IL-6	MBS	MBS2529848	1:1000	WB	Rabbit
	NF-κBp65	Cell signaling	#6956	1:1000	WB	Mouse
	COX-2	Cayman	160106	1:1000	WB	Rabbit
	MPO	Abclonal	A1374	1:1000	WB	Rabbit
	NE	Abcam	ab68672	1:1000	WB	Rabbit
Iron homeostasis	Ferritin	Abcam	ab75973	1:1000	WB	Rabbit
	LCN2	R&D	AF1857	1:1000, 1:200	WB, IF	Goat
	24p3R	ProSci	4651	1:1000, 1:200	WB, IF	Rabbit
	Celuroplasmin	Abcam	ab48614	1:1000	WB	Rabbit
Oxidative stress	iNOS	BD	610332	1:1000	WB	Rabbit
	HO-1	Enzo	ADI-SPA-895	1:3000, 1:200	WB, IF	Rabbit
Loading control	IgG	Santa Cruz	sc-52336	1:1000	WB	Mouse
	p84	Abcam	ab487	1:3000	WB	Mouse
	β-actin	Sigma	A5441	1:50,000	WB	Mouse

WB, Western blot; IF, immunofluorescence.

## Data Availability

Not applicable.

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
