# Peer review of "Lipocalin-2 Deficiency Reduces Oxidative Stress and Neuroinflammation and Results in Attenuation of Kainic Acid-Induced Hippocampal Cell Death"

_antioxidants, 2021, doi:10.3390/antiox10010100_

Round 1

Reviewer 1 Report

This paper indicated that the effects of LCN2 deficiency on kainic acid-induced cell death in the mouse hippocampus by inspecting various factors. Data are of interest and may contribute to future studies to clarify underlying mechanisms of neuronal cell death and to explore clinical treatments to prevent brain damages. However, there are several concerns that the authors should address before publication.

  1. Title: Based on the text, this paper focused on not only the cell death, but also the various factors that may result in the cell death. The present title is not reflected the latter part. The title, for example, “Lipocalin-2 deficiency reduced oxidative stress and neuroinfammation markers and resulted in attenuation of kainic acid-induced hippocampal cell death” may be better if the number of characters is allowed.
  2. Line 19: It is not clear that the animals in which serum LCN2 was increased by KA injection are wild type, or knockout mice, or both.
  3. It is very difficult to follow the text. The main reason for this may be the purpose of each investigating factor and their relationship are not clear. It is better to summarize all investigated factors indicating their purposes in Line 50 and 51 without abbreviations. Alternatively, add each purpose at the end of subheading, for example, “2.3. Tissue preparation and cresyl violet staining to inspect general histology and pyknotic nuclei”. The latter may be better than the former. Or, one more another way, this is the best, is to make a schematic drawing to clarify the relationship among all factors which the authors inspected.
  4. The hippocampus is not simple structures. Western blotting analyses may be used the whole hippocampus including the dentate gyrus, however, histological investigation pointed about sub-structures of the hippocampus. For example, TUNEL staining was seen in the CA3 and Perls’s staining was observed in the CA1. Focusing levels between these two techniques are different, but data are concluded as a same level. This reviewer knows there are technical limitations (when use a stereoscopic microscope, it is possible to separate each sub-structures of the hippocampus), however, at least, the authors should note whether other sub-structures have positive cells or not in histological inspection.
  5. The authors clarified changes of various markers. However, it may imply miscellaneous. This reviewer recommends summarizing results by a table and categorizing the various markers, if possible.
  6. Line 64: “8” should be “Eight”.
  7. Line 169: There is a starting “[“, but no ending”]”.
  8. Lines 288 and 289: When LCN2 is synthesized by other peripheral cells, it is not certain that all circulating LCN2 is secreted from brain cells because kainic acid is injected intraperitoneally.

Author Response

  1. Title: Based on the text, this paper focused on not only the cell death, but also the various factors that may result in the cell death. The present title is not reflected the latter part. The title, for example, “Lipocalin-2 deficiency reduced oxidative stress and neuroinflammation markers and resulted in attenuation of kainic acid-induced hippocampal cell death” may be better if the number of characters is allowed.

→ As the reviewer suggested, we corrected it.

  1. Line 19: It is not clear that the animals in which serum LCN2 was increased by KA injection are wild type, or knockout mice, or both.

→ In LCN2 knockout mice, no detection of serum LCN2 is available. We corrected it as follows; Circulating LCN2 levels were significantly increased in KA-treated wild-type (WT) mice.

  1. It is very difficult to follow the text. The main reason for this may be the purpose of each investigating factor and their relationship are not clear. It is better to summarize all investigated factors indicating their purposes in Line 50 and 51 without abbreviations. Alternatively, add each purpose at the end of subheading, for example, “2.3. Tissue preparation and cresyl violet staining to inspect general histology and pyknotic nuclei”. The latter may be better than the former. Or, one more another way, this is the best, is to make a schematic drawing to clarify the relationship among all factors which the authors inspected.

→ We revised this sentence as follows; To inspect general histology and pyknotic nucleus, the sections were stained with Cresyl violet,

  1. The hippocampus is not simple structures. Western blotting analyses may be used the whole hippocampus including the dentate gyrus, however, histological investigation pointed about sub-structures of the hippocampus. For example, TUNEL staining was seen in the CA3 and Perls’s staining was observed in the CA1. Focusing levels between these two techniques are different, but data are concluded as a same level. This reviewer knows there are technical limitations (when use a stereoscopic microscope, it is possible to separate each sub-structures of the hippocampus), however, at least, the authors should note whether other sub-structures have positive cells or not in histological inspection.

→ As the reviewer commented, we again checked Perls’s stained slides. However we found that CA1 is a typo and collect it to CA3.

As like our findings, because kainate receptors (KARs) primarily are most abundant in CA3 region, all histological investigations in CA3 regions have been observed. So, KA treatment causes hippocampal damage that occurred primarily in the CA3 region. It also reported that hippocampal blood flow compensates for the oxygen demand of the vulnerable CA3 regions during KA-induced seizure (1).

Reference>

  1. Pinard E, Tremblay E, Ben-Ari Y, Seylaz J. Blood flow compensates oxygen demand in the vulnerable CA3 region of the hippocampus during kainate-induced seizures. Neuroscience. 1984 Dec;13(4):1039-49.
  2. The authors clarified changes of various markers. However, it may imply miscellaneous. This reviewer recommends summarizing results by a table and categorizing the various markers, if possible.

→ As the reviewer suggested, we revised table 1.

  1. Line 64: “8” should be “Eight”.

→ We corrected it.

  1. Line 169: There is a starting “[“, but no ending”]”.

→ We corrected it.

  1. Lines 288 and 289: When LCN2 is synthesized by other peripheral cells, it is not certain that all circulating LCN2 is secreted from brain cells because kainic acid is injected intraperitoneally.

We thoroughly appreciate the reviewer’s valuable comment.

Kainic acid (kainate) is a neuroexcitotoxic and epileptogenic substance by acting on kainate receptors (KARs) in the brain. In particular, systemic KA treatment has been shown to increase of apoptosis in hippocampal neurons. Almost all data studies in KARs have been carried out on brain. However, in some studies the presence of kainite receptors was found only in the heart and retina (2,3). Szaroma et al. (4) reported that KA not only is toxic to the brain but also for the liver and kidneys of mice after intraperitoneal KA injection. It has been shown that KA causes the decrease in antioxidant activity in the brain as well as the liver and kidney 24 h after KA treatment. Therefore, as the reviewer commented, we agree that increase in circulating LCN2 level not only could be found in the hippocampus but also in peripheral organs. However, our findings indicate that LCN2-positive proteins in the third ventricle correlate both circulating and hippocampal LCN2 levels. In the present study, we suggest that secreted LCN2 levels could be predominantly appeared from KA-induced damaged CA3 region.    

Reference>

  1. DeVries SH, Schwartz EA. Kainate receptors mediate synaptic transmission between cones and 'Off' bipolar cells in a mammalian retina. Nature. 1999 Jan 14;397(6715):157-60.
  2. Wang LG, Zeng J, Yuan WJ, Su DF, Wang WZ. Comparative study of NMDA and AMPA/kainate receptors involved in cardiovascular inhibition produced by imidazoline-like drugs in anaesthetized rats. Exp Physiol. 2007 Sep;92(5):849-58.
  3. Szaroma W, Dziubek K, Greń A, Kreczmer B, Kapusta E. Influence of the kainic acid on antioxidant status in the brain, liver and kidneys of the mouse. Acta Physiol Hung. 2012 Dec;99(4):447-59.

Reviewer 2 Report

In this mauscript Shin et collegues examined the effect of genetic blocking Lipocalin-2 (LCN2 ko) on neuroinflammation and oxidative stress in KA-induced neuronal death.

Authors observed that LCN2 deficiency reduced neuronal cell death and BBB leakage in the KA-treated hippocampus.

Analysing LCN2 knockout mice, they found that the expressions of neutrophil markers myeloperoxidase and neutrophil elastase were decreased compared to their expressions in wild-type mice following KA treatment.

Furthermore, LCN2 deficiency also attenuated KA-induced iron overload and oxidative stress in the hippocampus.

Authors conclude that the presented results suggest that LCN2 may play an important role in iron-related oxidative stress and neuroinflammation in KA-induced hippocampal cell death.

In my opinion the general quality of the manuscript is accettable, however the analysis related the oxidative stress looks weak.

Considering the main focus of this journal and the enunciated conclusions line 315-316 : “In conclusion, our findings suggest that the genetic blocking of LCN2 has a neuroprotective effect against KA-induced neuronal death through a reduction in iron-related oxidative stress” I think that authors should put more effort in the confirmation of tha data related to oxidative stress i.e. what about SOD2 and ACO2? Could authors measure  ROS levels with a different approch in order to better support their conclusions ?

Minor:

I suggest to do another round of English editing and tipos revision:

line 33: Fe2+,    correct the comma

line 64: 8 weeks of WT and LCN2 (−/−) mice were used for this study. (to add old)

line 233: circulation LCN2….should be circulating LCN2

Fig. 1A add indicative arrows to the pictures

Author Response

  1. Considering the main focus of this journal and the enunciated conclusions line 315-316 : “In conclusion, our findings suggest that the genetic blocking of LCN2 has a neuroprotective effect against KA-induced neuronal death through a reduction in iron-related oxidative stress” I think that authors should put more effort in the confirmation of the data related to oxidative stress i.e. what about SOD2 and ACO2? Could authors measure ROS levels with a different approach in order to better support their conclusions?

→ Thanks for your critical comment. As the reviewer suggested, we additionally performed western blot analysis for SOD-1 and SOD-2 antibodies. However, we found that hippocampal SOD-1 and SOD-2 proteins were not significantly altered in KA-treated LCN2 KO mice compared with KA-treated WT mice. There were slightly decreased expressions of SOD-1 and SOD-2 proteins 24 h after KA treatment compared to control mice. Maybe, according with their function in antioxidants, we propose that KA treatment may reduce the activity of antioxidants and lead to increased oxidative stress. So, we suggest that further experiments on the effects of LCN2 deficiency on antioxidant enzymes are needed in the future.

Minor:

I suggest to do another round of English editing and typos revision:

  1. line 33: Fe2+,  correct the comma

→ We corrected it.

  1. line 64: 8 weeks of WT and LCN2 (−/−) mice were used for this study. (to add old)

→ We corrected it to Eight weeks old.

  1. line 233: circulation LCN2….should be circulating LCN2

→ We corrected it.

  1. Fig. 1A add indicative arrows to the pictures

→ As the reviewer suggested, we added it and revised as follows; Arrows indicate CA3 regions that appear with pyknotic nuclei.

Reviewer 3 Report

It is a very interesting manuscript where the damaging effect of the protein LNC2 on the hippocampus is demonstrated. Likewise, they demonstrate the neuroprotective effect of the LCN2 deletion in the LNC2 (- / -) mouse on oxidative stress and neuronal death. Emphasize that it is a suitable manuscript for publication. Only in the introduction should be discussed, other key targets in the process of neuronal death by KA / temporal lobe epilepsy such as JNK1 and JNK3. In the manuscript it should be included the articles in the references section : Role of c-Jun N-Terminal Kinases (JNKs) in Epilepsy and Metabolic Cognitive Impairment. Busquets O, Ettcheto M, Cano A, R Manzine P, Sánchez-Lopez E, Espinosa-Jiménez T, Verdaguer E, Dario Castro-Torres R, Beas-Zarate C, X Sureda F, Olloquequi J, Auladell C, Folch J, Camins A. Int J Mol Sci. 2019 Dec 30; 21 (1): 255 JNK Isoforms Are Involved in the Control of Adult Hippocampal Neurogenesis in Mice, Both in Physiological Conditions and in an Experimental Model of Temporal Lobe Epilepsy. Castro-Torres RD, Landa J, Rabaza M, Busquets O, Olloquequi J, Ettcheto M, Beas-Zarate C, Folch J, Camins A, Auladell C, Verdaguer E. Mol Neurobiol. 2019 Aug; 56 (8): 5856-5865.

Author Response

Only in the introduction should be discussed, other key targets in the process of neuronal death by KA / temporal lobe epilepsy such as JNK1 and JNK3. In the manuscript it should be included the articles in the references section : Role of c-Jun N-Terminal Kinases (JNKs) in Epilepsy and Metabolic Cognitive Impairment. Busquets O, Ettcheto M, Cano A, R Manzine P, Sánchez-Lopez E, Espinosa-Jiménez T, Verdaguer E, Dario Castro-Torres R, Beas-Zarate C, X Sureda F, Olloquequi J, Auladell C, Folch J, Camins A. Int J Mol Sci. 2019 Dec 30; 21 (1): 255 JNK Isoforms Are Involved in the Control of Adult Hippocampal Neurogenesis in Mice, Both in Physiological Conditions and in an Experimental Model of Temporal Lobe Epilepsy. Castro-Torres RD, Landa J, Rabaza M, Busquets O, Olloquequi J, Ettcheto M, Beas-Zarate C, Folch J, Camins A, Auladell C, Verdaguer E. Mol Neurobiol. 2019 Aug; 56 (8): 5856-5865.

→ As the reviewer suggested, we added one sentence and two references in the introduction section as follows (line 39-41); Moreover, previous studies have shown that c-Jun N-terminal kinase-signaling pathway plays an important role in the process of neuronal death in experimental epilepsy model [9,10].
